# Transgene removal using an *in cis* programmed homing endonuclease via single-strand annealing in the mosquito *Aedes aegypti*
Keun Chae, Bryan Contreras, Joseph S. Romanowski, Chanell Dawson, Kevin M. Myles & Zach N. Adelman ✉

While gene drive strategies have been proposed to aid in the control of mosquito-borne diseases, additional genome engineering technologies may be required to establish a defined end-of-product-life timeline. We previously demonstrated that single-strand annealing (SSA) was sufficient to program the scarless elimination of a transgene while restoring a disrupted gene in the disease vector mosquito *Aedes aegypti*. Here, we extend these findings by establishing that complete transgene removal (four gene cassettes comprising ~8-kb) can be programmed *in cis*. Reducing the length of the direct repeat from 700-bp to 200-bp reduces, but does not eliminate, SSA activity. In contrast, increasing direct repeat length to 1.5-kb does not increase SSA rates, suggesting diminishing returns above a certain threshold size. Finally, we show that while the homing endonuclease Y2-I-*Ani*I triggered both SSA and NHEJ at significantly higher rates than I-*Sce*I at one genomic locus (*P5-EGFP*), repair events are heavily skewed towards NHEJ at another locus (*kmo*), suggesting the nuclease used and the genomic region targeted have a substantial influence on repair outcomes. Taken together, this work establishes the feasibility of engineering temporary transgenes in disease vector mosquitoes, while providing critical details concerning important operational parameters.

*Aedes aegypti* is a highly invasive mosquito vector that transmits arboviral pathogens such as Zika, yellow fever, dengue, and chikungunya viruses, representing a substantial threat to public health in developing countries in tropical and sub-tropical regions[1–3]. Due to the scarcity of highly effective, affordable vaccines or antiviral treatments and the mosquito's increasing resistance to pesticides, the development of genetics-based control strategies, which include the generation of genetically engineered (GM) mosquitoes and their release to natural habitats, has been highlighted as a fundamentally preventive tool for mosquito-borne diseases[4–6]. Genetic control strategies such as the Release of Insects carrying a Dominant Lethal (RIDL) and gene drive have been proposed to reduce the wild species or inactivate their disease-transmitting ability [reviewed in refs. 7,8]. In particular, the emergence of CRISPR-based gene editing[9] has significantly accelerated technical advances in homing gene drive approaches in *Anopheles* and *Aedes* mosquitoes that can rapidly spread beneficial transgenes into laboratory populations[10–20].

While CRISPR-based homing gene drive is a potentially powerful tool for efficiently altering the genomes of disease vectors for the purposes of either population suppression or population replacement[21,22], its anticipated highly invasive, self-propagating nature has also raised public concerns of unforeseen ecological risks[23–27]. This restriction has led to the development of gene drive with local confinement[17], self-exhaustion[28], and anti-CRISPR breaks such as CATCHA, e-CHACRs, ERACRs, and the anti-CRISPR AcrIIA4 protein[29–31]. We previously conceptualized a model for erasing gene drive transgenes from the genome and reversing the invasion back to the wild-type state[32]. Such a transgene elimination system was successfully engineered in the *Aedes aegypti* genome[33], and was based on single-strand annealing (SSA), a eukaryotic DNA repair mechanism that triggers a deletion when a double-strand break (DSBs) occurs between two identical sequence motifs, known as direct repeats (DRs)[34,35].

In this study, we demonstrate that transgene elimination and restoration of wild-type alleles can be triggered *in cis*, as well as in trans, an important next step in constructing a truly self-eliminating transgene.

Department of Entomology, Texas A&M University, College Station, TX 77843, USA. ✉e-mail: zachadel@tamu.edu

We also evaluated several parameters of the SSA mechanism for transgene elimination efficiency in *Ae. aegypti*. For this, we generated transgenic mosquito strains featuring various changes predicted to affect rates of SSA, such as the lengths of the DR sequence (0.2-kb; 0.7-kb; 1.5-kb), the distances between the DRs (spacer sizes: 3.7-kb; 7.9-kb), and the types of the homing endonuclease (I-*Sce*I; Y2-I-*Ani*I). Our organismal DNA repair assays reveal that various size ratios of Cargo-to-DR are feasible in the *Ae. aegypti* genome with context-dependent self-eliminating dynamics.

## Results
### Lengths of DR sequences can influence the rate of SSA in regard to removing transgenic cargo from the *Aedes aegypti* genome

We previously demonstrated effective SSA utilizing a 0.7-kb direct repeat flanking two transgene cassettes[33]. To determine the influence of direct repeat (DR) length on the effectiveness of SSA, we generated a parallel transgenic strain at the same insertion site with a 0.2-kb DR and compared it to our previous construct (0.7-kb) in regard to SSA-mediated elimination of transgene components containing *DsRED* and *EGFP* (3.7-kb) integrated to exon 4 of *Ae. aegypti kmo* (Fig. 1a). In both cases, transgenic mosquitoes have marker phenotypes of white (Kmo⁻) eyes, DsRED⁺ eyes, and EGFP⁺ body, and the recognition sequence of the homing endonuclease I-*Sce*I

positioned in-frame next to the ATG translational start codon of *DsRED*. For 0.7-kb and 0.2-kb of DRs, *kmo* exons 2/3/4 and exons 3/4 were engineered in the cassettes, respectively (Fig. 1a, yellow bars), which created duplicates of the chromosomal counterparts (Fig. 1a, pink bars). Correct integration at the *kmo* locus was verified by PCR analysis using genomic DNAs obtained from *kmo*^DR0.2-SceI^ and *kmo*^DR0.7-SceI^ (Fig. 1b; Supplementary Data 1, Table S1).

To determine the effect of DR length variation on transgene elimination, we performed the SSA test in a split system, where DR-flanked transgenes and the I-*Sce*I nuclease (the SSA trigger) were provided from independent strains (Fig. 1c; Supplementary Data 1–2). F₁ offspring females carrying both SSA components were outcrossed with *kmo*^−/−^ males in triplicated experiments and allowed to individually oviposit. DNA repair events were scored in F₂ progeny for each female. Following I-*Sce*I DSB induction, NHEJ-driven indel mutations could frameshift the *DsRED* coding region, resulting in the phenotype WG (Kmo⁻; DsRED⁻; EGFP⁺). In contrast, repair by the SSA pathway would result in black eyes with no marker gene (Blk: Kmo⁺; DsRED⁻; EGFP⁻). These phenotypes were scored as: (i) the percentage of F₁ females that produced at least one progeny with an NHEJ or SSA phenotype (Fig. 1d), (ii) the percent of F₂ individuals per F₁ mother that displayed an NHEJ or SSA phenotype (Fig. 1e).

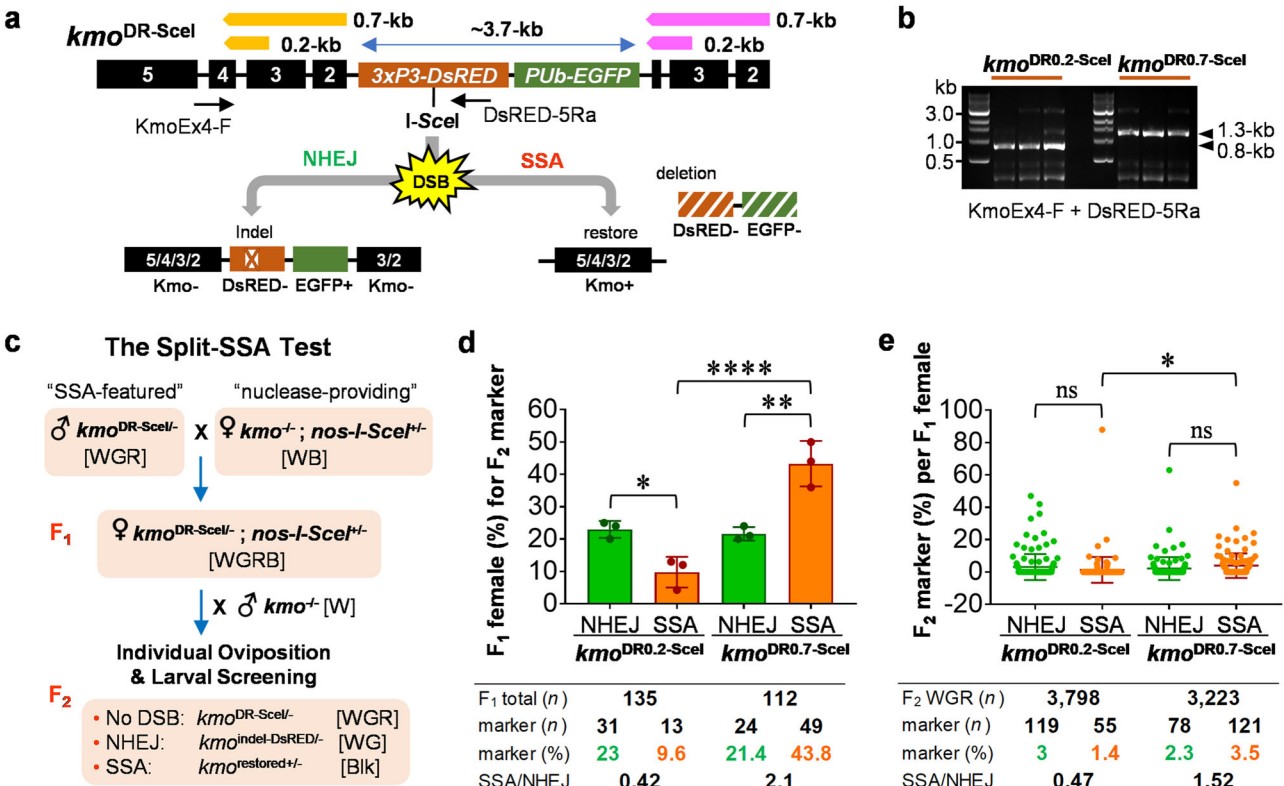

**Fig. 1 | Effects of direct repeat lengths on DNA repair outcomes at the *kmo* locus. a** Schematic representation of pSSA-KmoDR0.2-SceI and pSSA-KmoDR0.7-SceI transgenes integrated into the *Ae. aegypti* genome resulting in *kmo*^DR0.2-SceI^ and *kmo*^DR0.7-SceI^ strains, respectively. Direct repeat (DR) sequences of *kmo* exons 3/2 (yellow bars) and chromosomal counterparts (pink bars); the I-*Sce*I-induced DSB site at *DsRED* used to trigger the SSA pathway is indicated, along with locations of PCR primers utilized to confirm insertions (arrows). The DSBs induced at the I-*Sce*I site are processed through NHEJ to disrupt *DsRED* (Kmo⁻; DsRED⁻; EGFP⁺) or SSA to restore *kmo* with the deletion of transgenes (Kmo⁺; DsRED⁻; EGFP⁻). **b** PCR-based verification of insertion site for each transgenic mosquito strain. **c** The schematic workflow of the SSA test in a split system, where *kmo*^DR0.2-SceI^ or *kmo*^DR0.7-SceI^ male mosquitoes were crossed with nuclease-providing females. F₁ offspring females were outcrossed with *kmo*^−/−^ males and individually oviposited in a 24-well tissue culture plate with 2% low melting point agarose. F₂ larvae per female were scored for DNA repair event-dependent marker phenotypes. W, white eyes; G, EGFP⁺ body; R, DsRED⁺

eyes; Blk, black eyes. **d** DNA repair events were scored for F₁ female founders (triplicate groups). Each bar represents percentage of F₁ females that produced at least one NHEJ (WG, green bar) or SSA (Blk, orange bar) progeny. [F₁ total (n)], the total number of F₁ female founders; [marker (n or %)], the number or percentage of all F₁ females that produced at least one NHEJ or SSA progeny; [SSA/NHEJ], the ratio of [marker (%)] (note that founders that produced both types of events would be counted in both). Tukey's multiple comparisons test (1-way ANOVA): *P < 0.05; **P < 0.01; ****P < 0.0001. **e** The percentage of F₂ offspring per individual F₁ female for each DNA repair pathway-dependent marker phenotypes: WG for NHEJ [% WG/(WGR+WG+Blk)]; Blk for SSA [% Blk/(WGR+WG+Blk)]. Each dot represents rates of NHEJ and SSA events recovered in the progeny of a single F₁ female. [F₂ WGR (n)], the total number of F₂ WGR mosquitoes; [marker (n or %)], the number or percentage of F₂ progenies that have the NHEJ (green) or SSA (orange) phenotype; [SSA/NHEJ], the proportional ratio of [marker (%)]. Tukey's multiple comparisons test (1-way ANOVA): *P < 0.05; ns not significant.

Both $kmo^{DR0.2-SceI}$ and $kmo^{DR0.7-SceI}$ had similar rates of NHEJ-mediated DSB repair when considering either $F_1$ mothers (~20%) or the percentage of total $F_2$ offspring (~2.5%). In contrast, $kmo^{DR0.2-SceI}$ showed significantly decreased rates of SSA (only 9.6% of $F_1$ mothers and 1.4% of total $F_2$ offspring), compared to $kmo^{DR0.7-SceI}$ (>40% of $F_1$ mothers and 3.5% of $F_2$ offspring). The ratio of SSA-to-NHEJ of $kmo^{DR0.2-SceI}$ was ~0.4, indicating that the DNA repair pathway selection was skewed toward NHEJ, while those of $kmo^{DR0.7-SceI}$ was ~1.5–2. We conclude that while 0.2-kb of DR was still capable of triggering SSA and could eliminate 3.7-kb of transgenic cargo, it was less efficient than the longer 0.7-kb repeat.

## SSA-based transgene elimination is effective *in cis* as well as in trans

While we have demonstrated that SSA-mediated transgene elimination is possible in a split system, in order to provide temporal control over an invasive gene drive element it must also be effective *in cis*. That is, the source of the nuclease must be encoded within the DRs and would also be removed upon successful SSA. To determine if this was possible, we developed two additional integrations at the *kmo* locus consisting of the 1 piece-SSA transgene (7.9-kb), in which the homing endonuclease I-*Sce*I gene (the SSA trigger) was engineered together with its target site and the DRs flanking a series of marker genes (Fig. 2a). These two SSA strains were designed to have 0.7-kb of *kmo* exons 2/3/4 or 1.5-kb of *kmo* exons 2/3/4 and intron 1 as the DR motifs, respectively for $kmo^{DR0.7-nos-SceI}$ and $kmo^{DR1.5-nos-SceI}$ (Supplementary Table S1). Both transgenic cargos include eye-specific *DsRED* with the I-*Sce*I target site, eye-specific *BFP*, whole body *EGFP*, and a germline-specific *nos* promoter-controlled *I-SceI*. PCR analysis using genomic DNA and various primer pairs (Fig. 2a; Supplementary Table S2) verified correct integration of each transgene, including the size difference (0.8-kb) between the two transgene versions, the presence of *nos-I-SceI*, and chromosomal integration of transgenes to the *kmo* gene (Fig. 2b).

To analyze DNA repair outcomes, transgenic females (WGRB: Kmo⁻; EGFP⁺; DsRED⁺; BFP⁺) were outcrossed with $kmo^{-/-}$ males (W: Kmo⁻) in triplicate, and DNA repair pathway-dependent marker phenotypes were scored in offspring larvae for NHEJ (WGB: Kmo⁻; EGFP⁺; DsRED⁻; BFP⁺) and SSA (Blk: Kmo⁺; EGFP⁻; DsRED⁻; BFP⁻) (Fig. 2c; Supplementary Tables S3–S5, Supplementary Data 3). For each generation ($G_8$, $G_9$, $G_{10}$), WGRB females (n = 25 per replicate) were screened by high-resolution melt analysis (HRMA) to eliminate any silent/in-frame indel mutation that occurred at the I-*Sce*I target site, and only mosquitoes with the intact transgene were taken to be founding mothers for the next generation ($F_{n+1}$) (Supplementary Fig. S1). Both strains, $kmo^{DR0.7-nos-SceI}$ and $kmo^{DR1.5-nos-SceI}$, were shown to maintain similar rates of SSA- or NHEJ-mediated repair for nuclease-induced DSBs throughout multiple generations (Fig. 2c). Similarly, the proportion of $G_{10}$ females that produced at least one NHEJ or SSA event was not statistically different between the two strains (Fig. 2d; Supplementary Data 4), nor was the percentage of total progeny at $G_{11}$ that exhibited NHEJ or SSA phenotypes (Fig. 2e; Supplementary Data 5) despite the appearance of a trend towards higher rates of SSA in the longer $kmo^{DR1.5-nos-SceI}$ strain (15.2% of mothers in $G_{10}$ and 1.8% of progeny in $G_{11}$) compared to those of $kmo^{DR0.7-nos-SceI}$ (5.3% of mothers at $G_{10}$ and 0.6% of progeny in $G_{11}$). Together, these results confirm that SSA-based transgene elimination can be programmed *in cis*, as well as in trans. Rates of SSA appeared lower than the in trans experiment, a phenomenon that may be due to the change in transgene configuration or simply the expanded distance between the DRs.

Unexpectedly, we found some mosquitoes displayed eye-specific EGFP fluorescence (WG-eye: Kmo⁻; EGFP⁺; DsRED⁻; BFP⁻) (Fig. 2f), which might have resulted from an aberrant, unidentified process of DNA repair. PCR analysis using the primer pair of Hsp70-F and GFP-3R in these individuals showed amplicons of 1.3-kb and 1.6-kb, smaller (about half) than predicted (Fig. 2a, >3-kb). Sequence analysis revealed that a range of sequences pertaining to [DsRED and SV40] and [downstream parts of the *PUb* promoter] were deleted, placing the EGFP gene close enough to be expressed by the eye-specific 3xP3 promoter. This aberrant conversion

event was recovered at a frequency of ~0.1% in our assay (n = 6, out of >6000 WGRB mosquitoes).

## Choice of homing endonuclease strongly impacts the efficiency of SSA DSB repair in *Aedes aegypti*

Previously, we determined that the LAGLIDADG-type homing endonucleases I-*Ani*I, I-*Cre*I, and I-*Sce*I are effective at inducing DSBs for genome editing in *Ae. aegypti*[36] when expressed transiently following embryonic injection. While I-*Sce*I was successfully engineered to induce germline-specific DSBs to trigger SSA-based self-elimination of a *kmo*-integrated transgenic cargo[33], I-*Ani*I was actually much more efficient in our previous experiments[36]. To determine the impact of nuclease choice on DSB repair and SSA efficiency, we generated a new DSB trigger strain, *nos-I-AniI*, which expresses Y2-I-*Ani*I[37] (I-*Ani*I, here in after) under the control of germline-specific *nos* promoter and 3′UTR (Supplementary Tables S1 and S6) and tested its activity to trigger DSB repair processing in the transgene of the *Mariner Mos1*-driven transgenic *P5-EGFP* strain[36]. In *P5-EGFP*, there are two groups of homing endonuclease target sites [sites-A] and [sites-B] and two gene cassettes [3xP3-DsRED-SV40] and [PUb-EGFP-SV40] (Fig. 3a). DSBs can trigger SSA by diverse pairs of DR elements such as [SV40], [loxP], and [individual HE sites] and are expected to remove [PUb-EGFP-SV40] from the transgene, resulting in the WR phenotype (Kmo⁻; EGFP⁻; DsRED⁺).

The *P5-EGFP* strain was reciprocally crossed with either *nos-I-AniI* or *nos-I-SceI* strains in triplicate, and DNA repair pathway-dependent marker phenotypes were scored in $F_2$ offspring by each nuclease-providing lineage (Fig. 3b; Supplementary Table S7, Supplementary Data 6). Rates of SSA (WR phenotype) were substantially higher with I-*Ani*I (~21%) than I-*Sce*I (~0.4%) in $F_2$ progeny from the female lineage. SSA events were substantially lower in the male-initiated lineages, likely due to the lack of maternally contributed *nos-I-AniI*. As NHEJ events were not directly scorable, we performed HRMA of PCR amplicons at [sites-A] using a primer pair of SV40-F and PUb-5R (shown in Fig. 3a) from $F_2$ WGR progeny from *nos-I-AniI* (n = 73) and *nos-I-SceI* individuals (n = 80) (Fig. 3c). In sequence analysis, NHEJ at [sites-A] was identified to occur ~25% for *nos-I-AniI*; no such events were recovered for *nos-I-SceI* (Fig. 3d).

As WR phenotypes could be due to either SSA or to NHEJ following DSB induction at both sites-A and sites-B, with multiple DR features present in *P5-EGFP*, we further determined the rates of individual DR motif-dependent SSA events using high-throughput amplicon sequencing (Fig. 4). Genomic DNA was obtained from triplicate groups of $F_2$ larvae with WR phenotypes (n = 100 per replicate) and utilized for PCR amplification of the processed region (Supplementary Fig. S2). The nanopore-based sequencing results (n = 794 reads, see Materials and Methods) were aligned with individual signature sequences (~60 to 100-bp, blue dotted lines in Fig. 4) of seven types of DNA repair end-products, which can be produced by diverse SSA- or NHEJ-mediated processes of DSB repair occurring at either [sites-A] or [sites-B] in *P5-EGFP*. SSA events were recovered most abundantly by SV40 DRs (>59%) (Fig. 4), potentially due to the larger size (~230-bp) of these sequences. Interestingly, just over 30% of events appear to be the result of SSA using the two I-*Cmo*I recognition sites as DRs, despite the small size (20-bp) of this sequence. Likewise, a small subset of SSA events occurred by loxP (2.5%) or I-*Sce*I (2.9%) target sequences, but not by I-*Cre*I or I-*Ppo*I target sites, despite a similar length (Fig. 4). Finally, ~3% of events centered on the I-*Ani*I site, and could have been generated either by SSA (following 1 DSB) or NHEJ (following 2 DSBs) (Fig. 4). Changing alignment sensitivity or performing de novo clustering of nanopore reads did not meaningfully affect the rates of each reported repair outcome (Supplementary Note 1).

We also noted that some of the possible repair outcomes we anticipated (Fig. 4) preserve an intact I-*Ani*I target site, and thus themselves could be cleaved and potentially subject to a new round of SSA/NHEJ. For example, an initial SSA event mediated by SV40 DRs or between I-*Ani*I target sites can be further subject to DSB induction for either NHEJ-driven indel formation or SSA-based repair via I-*Cmo*I, loxP, or I-*Sce*I DRs (Fig. 4).

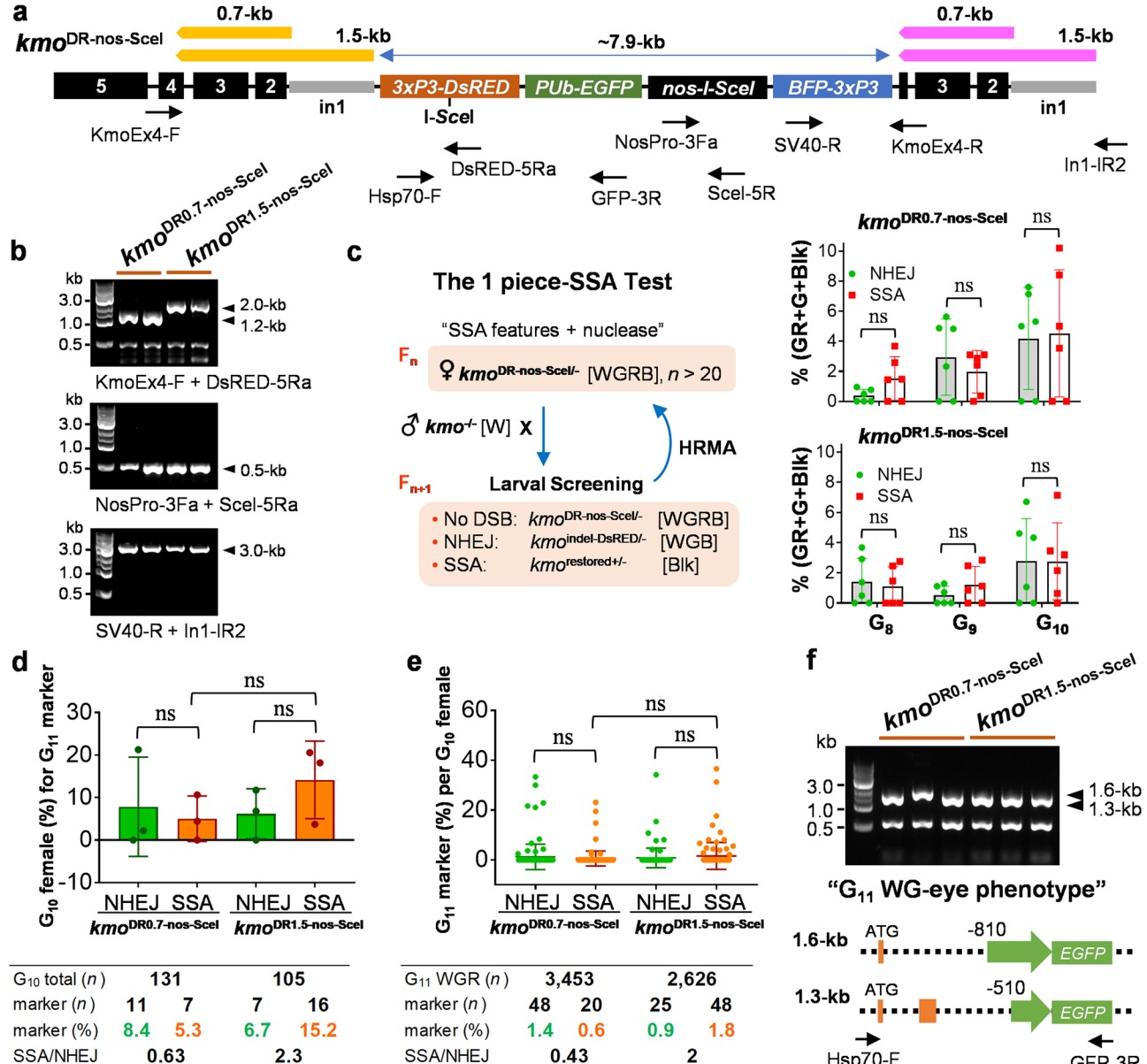

**Fig. 2 | SSA-based transgene elimination can be catalyzed *in cis*. a** Schematic representation of pSSA-KmoDR0.7-nos-SceI and pSSA-KmoDR1.5-nos-SceI integrated into genomes of $kmo^{DR0.7-nos-SceI}$ and $kmo^{DR1.5-nos-SceI}$ strains, respectively. The *kmo* exons 3/2 and exons 3/2-intron 1 sequences (yellow bars) were engineered for DR0.7 and DR1.5 motifs with chromosomal counterparts (pink bars), respectively. I-*Sce*I-target site is shown, with I-*Sce*I coding region under the control of the germline-specific *nos* gene promoter/3′UTR. **b** PCR-based verification of transgene insertion to verify insertion site, DR length gap between the two strains (KmoEx4-F + DsRED-5Ra), the presence of *nos-I-SceI* (NosPro-3Fa + SceI-5Ra), and the chromosomal integration of transgenes (SV40-R + In1-IR2). **c** The schematic workflow of the 1 piece-SSA test over multi-generations. For each generation, females ($n = 20–25$) carrying the intact transgene (WGR) as confirmed by HRMA were out-crossed with $kmo^{-/-}$ males. In the next generation, offspring larvae were scored for marker phenotypes to determine DNA repair events: W, white eyes; G, EGFP$^+$ body; R, DsRED$^+$ eyes; Blk, black eyes. Tukey's multiple comparisons test (1-way ANOVA): ns not significant. **d** Each bar represents percentage of G$_{10}$ females

that produced at least one NHEJ (WG, green bar) or SSA (Blk, orange bar) progeny from three independent experimental groups. [G$_{10}$ total ($n$)], the total number of G$_{10}$ female founders; [marker ($n$ or %)], the number or percentage of all G$_{10}$ females that produced at least one NHEJ or SSA progeny; [SSA/NHEJ], the ratio of [marker (%)] (note that founders that produced both types of events would be counted in both). Tukey's multiple comparisons test (1-way ANOVA): ns not significant. **e** Percent of G$_{11}$ offspring per individual G$_{10}$ female for DNA repair pathway-dependent marker phenotypes: WG for NHEJ [% WG/(WGR+WG+Blk)]; Blk for SSA [% Blk/(WGR+WG+Blk)]. Each dot represents rates of NHEJ and SSA events recovered in the progeny of a single G$_{10}$ female. [G$_{11}$ WGR ($n$)], the total number of G$_{11}$ WGR mosquitoes; [marker ($n$ or %)], the number or percentage of G$_{11}$ progenies that have the NHEJ (green) or SSA (orange) phenotype; [SSA/NHEJ], the proportional ratio of [marker (%)]. Tukey's multiple comparisons test (1-way ANOVA): ns not significant. **f** PCR analysis of genomic DNAs from white/EGFP$^+$ eyed-mosquitoes using indicated primers from G$_{11}$ mosquitoes. Identified large deletions are shown.

These secondary cleavage events are likely to be abundant, as about half ($n = 252$, 48.2%) of the SSA products obtained via DRs of SV40 ($n = 523$) (Fig. 4), were shown to contain indel mutations at the I-*Ani*I site (Supplementary Fig. S2). This might also explain the failure to obtain SSA events utilizing the I-*Ppo*I or I-*Cre*I recognition sites since both of those products

can be re-cleaved and further processed with additional SSA, whereas events utilizing loxP, I-*Sce*I or I-*Cmoe*I sequences cannot be further processed. Taken together, we conclude that the frequency of SSA repair within the *P5-EGFP* is strongly influenced by the nuclease used (I-*Sce*I vs. I-*Ani*I), and can be as high as 25% of the total progeny.

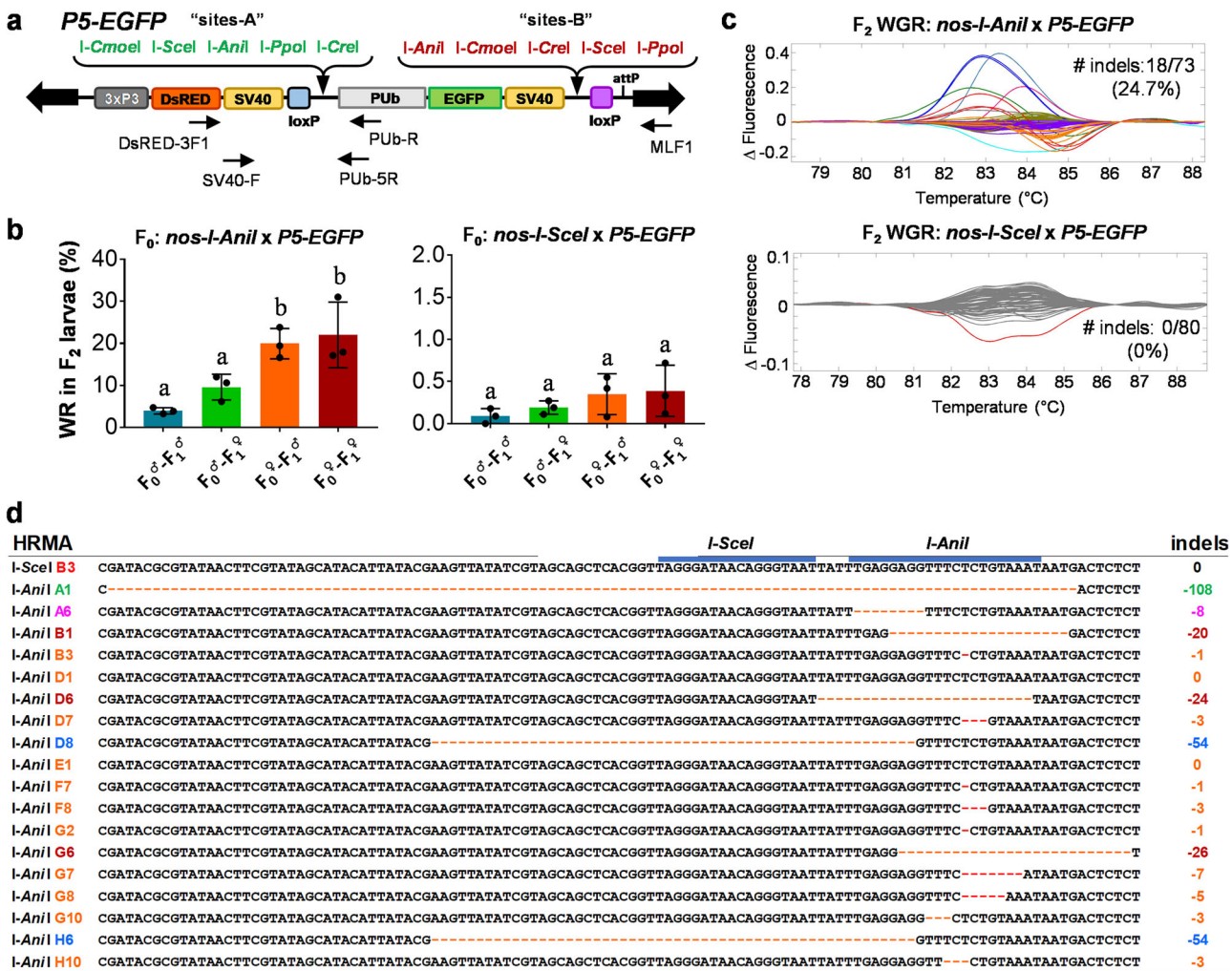

**Fig. 3 | Germline expression of I-*Ani*I induces substantially more SSA and NHEJ than I-*Sce*I in the *P5-EGFP* transgenic locus. a** Schematic representation of the SSA-capable *P5-EGFP* transgene integrated into the *Ae. aegypti* genome containing two marker genes (eye-specific *DsRED* and whole body-expressed EGFP) interspersed with two groups of homing endonuclease target sites (site-A and site-B) were engineered to flank the *EGFP* cargo. **b** Percentage of $F_2$ larvae with phenotypic loss of the *EGFP* marker from triplicate experiments. The X-axis indicates the grandparent ($F_0$)/parent ($F_1$) contributing the nuclease gene. Tukey's multiple comparison test

was found to be significant (2-way ANOVA, $P < 0.001$), and statistically different groups are marked (**a** and **b**). **c** HRMA of the site-A for $F_2$ mosquitoes scored as WGR (white and DsRED⁺ eyes; EGFP⁺ body) by using a primer pair of SV40-F and PUb-5R. **d** Sequence analysis of amplicons with altered melting curves from (**c**), compared to the baseline. Sequencing analysis revealed various numbers of nucleotide deletions occurred at site-A. The colors of IDs designate individual HRMA peaks shown in (**c**).

## I-*Ani*I-induced DSBs are almost exclusively repaired via NHEJ-mediated processes for a transgene engineered at the *kmo* locus

Based upon the substantially higher levels of SSA obtained with I-*Ani*I at the *P5-EGFP* locus, we attempted to maximize SSA-based elimination for the *kmo*-integrated transgene (~3.7-kb) by engineering 1.5-kb of DRs and the I-*Ani*I recognition site (Fig. 5a) in a similar manner to our previous constructs. PCR analysis using genomic DNAs obtained from *kmo*^DR0.7-SceI,33 and *kmo*^DR1.5-AniI (Supplementary Table S1) and the primer pair of KmoEx4-F and DsRED-5Ra verified integration site and the anticipated size difference (0.5-kb) between the two DR integrations (Fig. 5b). Subsequent sequencing analysis confirmed the two DSB sites, I-*Sce*I (5′-TAGGGATAA-CAGGGTAAT-3′) and I-*Ani*I (5′-TTGAGGAGGTTTCTCTGTAAATA-3′), which were engineered in-frame next to the ATG translational start codon of *DsRED* in *kmo*^DR0.7-SceI and *kmo*^DR1.5-AniI strains, respectively.

*kmo*^DR0.7-SceI or *kmo*^DR1.5-AniI males were crossed with the nuclease-providing females, *nos-I-SceI* or *nos-I-AniI* in triplicate. $F_1$ offspring females carrying both SSA components were outcrossed with *kmo*⁻/⁻ males. As before, females oviposited individually, and repair outcomes were scored within each family. For *kmo*^DR0.7-SceI, both NHEJ and SSA rates were similar

to each other in $F_1$ mothers (~25%, Fig. 5c; Supplementary Data 7) and $F_2$ offspring (1.6–2%, Fig. 5d; Supplementary Data 8), making the SSA/NHEJ ratios close to 1, similar to previous results[33], though slightly lower than that described in Fig. 1. In contrast, for *kmo*^DR1.5-AniI, 97% of $F_1$ mothers produced NHEJ-mediated DSB repair events, and ~50% of $F_2$ offspring larvae showed NHEJ phenotypes. Accordingly, total SSA rates of *kmo*^DR1.5-AniI were substantially lower than those of *kmo*^DR0.7-SceI, especially in $F_1$ mothers. Sequence analysis confirmed that WG phenotypes were indeed associated with indel mutations (−1, −2, and −4-bp) generated at the I-*Ani*I site (Supplementary Fig. S4). Overall, repair of I-*Ani*I-induced DSBs at the *Ae. aegypti kmo* locus was dramatically biased toward the NHEJ pathway rather than SSA, despite the increased length of DRs (1.5-kb).

## The SSA-based transgene removal technology and genetic control approaches

Summarizing all SSA tests we performed at the *kmo* locus showed that there was little correlation between the ratio of cargo/spacer to DR (*R*) in terms of elimination of *kmo*-integrated transgenes (Fig. 6). For the 3.7-kb-sized cargo, 0.2-kb and 0.7-kb of DRs were engineered for *R*18.5 and *R*5.3, respectively.

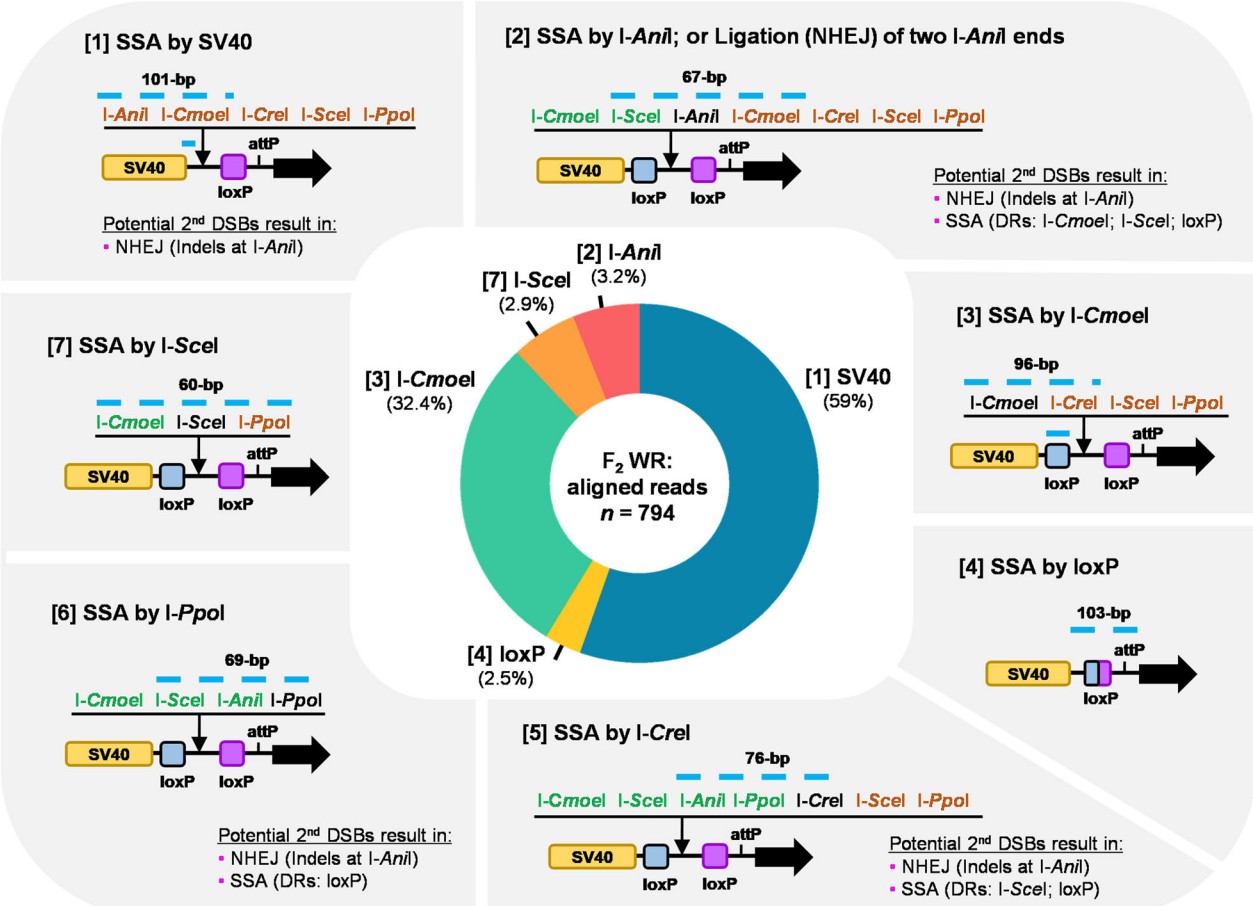

**Fig. 4 | Molecular characterization of SSA events triggered by I-AniI at the P5-EGFP transgene.** Predicted end-products resulting from DR-associated DNA repair processes and their rate of occurrence in individuals that lost the P5-EGFP transgene. Blue dotted lines indicate signature sequences that are partially unique to each end-product. 'Potential 2nd DSBs' describe additional SSA or NHEJ processes that are predicted to occur on the I-AniI site when it remains intact after the initial DSB repair event.

While R5.3 provided a relatively equal DNA repair pathway selection (SSA/NHEJ = 0.75 or 1.52), R18.5 resulted in a level of bias (SSA/NHEJ = 0.47). When the 7.9-kb-sized cargo was engineered with 0.7-kb (R11.3) or 1.5-kb (R5.3) of DRs, the average rate of SSA was higher in R5.3 (1.75%) than R11.3 (0.49%) while NHEJ rates were similar to each other. We conclude that the spacer length is relatively unimportant, while the DR length contributes only minimally to the selection of SSA. Both results are highly favorable to the development of SSA-based transgene removal approaches, allowing small repeats to potentially control large, complex transgene arrangements.

## Discussion

Nuclease-based gene editing technology relies on eukaryotic DNA repair mechanisms, and their efficiency can be dependent on how the locally induced DSBs are processed in the genome of the targeted species[38,39]. In eukaryotic genomes, DSBs are processed mainly by two distinct, antagonistic pathways: homology-directed repair (HDR) and non-homologous end joining (NHEJ)[40,41]. HDR enables the conversion of a homologous sister chromosome to the damaged allele, leading to error-free recovery of the DSB. A homing gene drive cassette (GD) induces DSBs at a specific genomic locus and subsequently triggers the conversion of heterozygosity (gene[GD/-]) to homozygosity (gene[GD/GD]) in germline cells, which forces the encoding gene to be passed on to most or all of the resulting offspring. In contrast, NHEJ directly ligates the broken DNA ends and does not require a template sequence, which often leads to sequence errors with nucleotide insertions or deletions (indels), especially being error-prone if DSBs would be repetitively triggered by prolonged nuclease activity[42]. Moreover, indels that occur at the

nuclease-targeted sequence may be resistant to further gene drive[43,44]. SSA is also a homology-based mechanism requiring resection of the DNA ends, but instead of a homologous chromosome or sister chromatid, it requires the recognition of identical sequences flanking the DSB site, which results in the elimination of the intervening segment of DNA[45]. Recently, we leveraged SSA to remove transgenic components and restore the function of the genetic target in the Aedes aegypti genome[33], and we found this activity was elevated by disruption of core NHEJ factors[46]. In this study, we further extend these findings by demonstrating that transgene removal can be catalyzed in cis, a necessary step towards building a fully self-eliminating gene drive. We also show that cargo of approximately 8-kb comprising 4 distinct gene cassettes can be removed via SSA. As genetic control approaches often require a large transgenic cargo including CRISPR/Cas9, gRNAs, and effectors, our data suggest that transgene insertion size alone is not likely to be a barrier for SSA-based approaches.

In regard to unforeseen risks of genetically modified organisms (GMOs), other transgene removal systems have been developed in plants. First, the Cre-loxP system was employed for removing the marker gene from transgenic plants[47]. Within the transgene, the neomycin phosphotransferase II (nptII) gene was flanked by two loxP sites. The heat-inducible activation of the Cre-recombinase selectively excised the nptII gene and eliminated kanamycin-resistance from the transgenic lines, while preserving the effector-driven trait. This system enabled the complete excision of the marker gene while retaining the scar of a loxP site the targeted gene. More recently, homology-directed repair (HDR) was engineered for marker-free, transgenic rice[48]. The encoding sequences for hygromycin

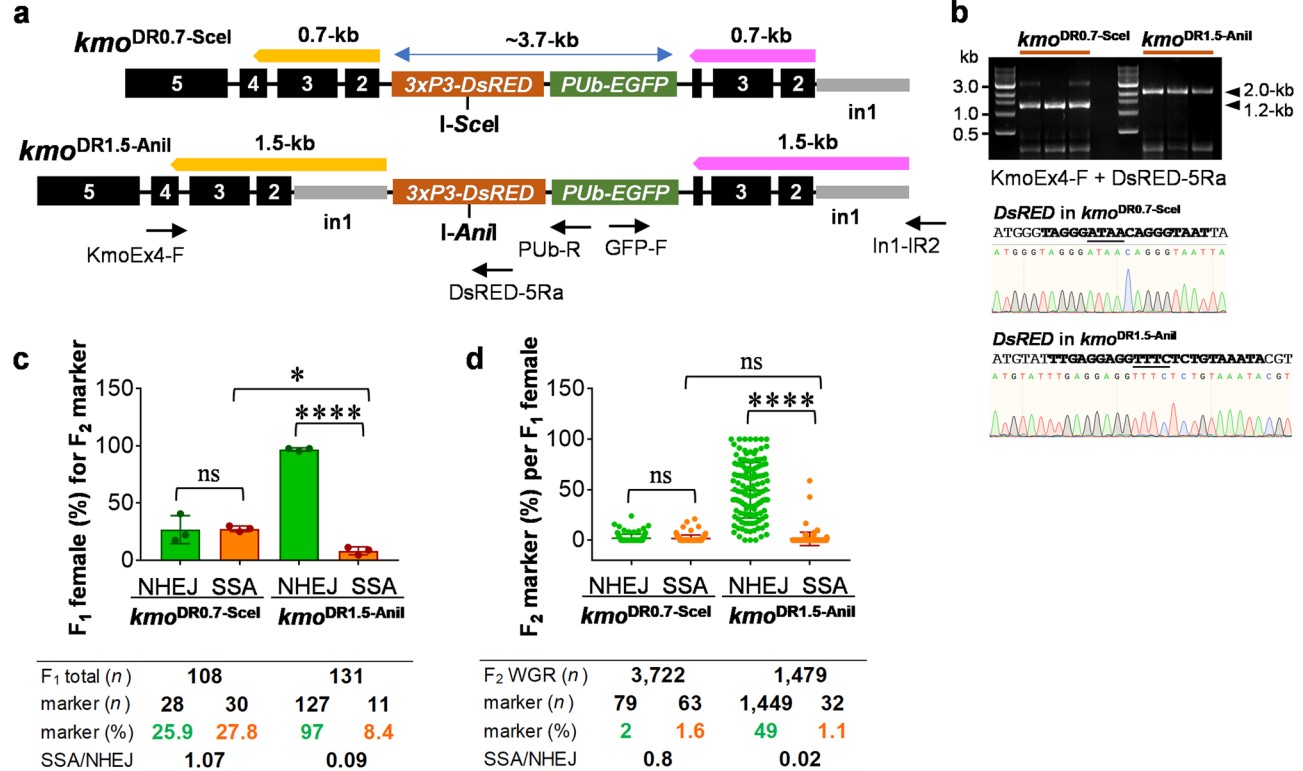

**Fig. 5 | Germline expressed Y2-I-*Ani*I predominately triggers NHEJ, and not SSA, at the *kmo* locus. a** Schematic representation of pSSA-KmoDR0.7-SceI and pSSA-KmoDR1.5-AniI integrated into genomes of *kmo*^DR0.7-SceI^ and *kmo*^DR1.5-SceI^ strains, respectively. The I-*Sce*I or Y2-I-*Ani*I target site was engineered at *DsRED* to trigger the SSA pathway. **b** PCR-based verification of transgenic mosquitoes. The PCR primer pair of KmoEx4-F and DsRED-5Ra was used to confirm the identity of the nuclease site engineered in each strain (Bold); The underlined sequence designates the overhang resulting from nuclease cleavage. **c** Each bar represents the proportion of $F_1$ females that produced at least one NHEJ (WG, green bar) or SSA (Blk, orange bar) progeny across three independent experiments. [$F_1$ total ($n$)], the total number of $F_1$ female founders; [marker ($n$ or %)], the number or percentage

of all $F_1$ females that produced at least one NHEJ or SSA progeny; [SSA/NHEJ], the ratio of [marker (%)] (note that founders that produced both types of events would be counted in both). Tukey's multiple comparisons test (1-way ANOVA): *$P < 0.05$; ****$P < 0.0001$; ns not significant. **d** Proportion of $F_2$ offspring exhibiting either NHEJ or SSA phenotypes per individual $F_1$ female: WG for NHEJ [% WG/ (WGR+WG+Blk)]; Blk for SSA [% Blk/(WGR+WG+Blk)]. Each dot represents rates of NHEJ and SSA events recovered in the progeny of a single $F_1$ female. [$F_2$ WGR ($n$)], the total number of $F_2$ WGR mosquitoes; [marker ($n$ or %)], the number or percentage of $F_2$ progenies that have the NHEJ (green) or SSA (orange) phenotype; [SSA/NHEJ], the proportional ratio of [marker (%)]. Tukey's multiple comparisons test (1-way ANOVA): ***$P < 0.0001$; ns not significant.

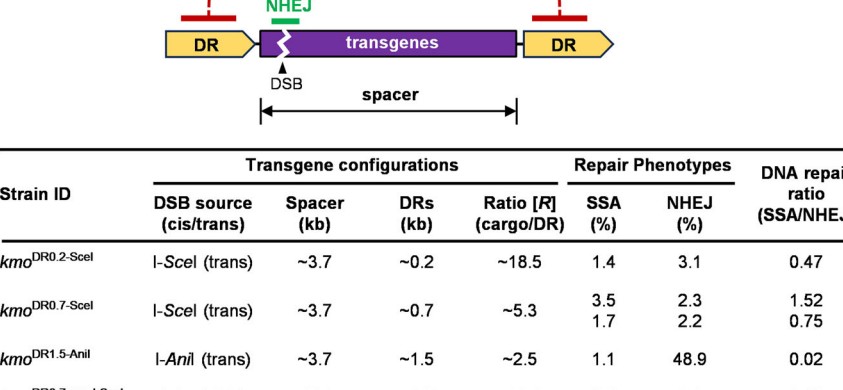

| Strain ID | Transgene configurations | | | | Repair Phenotypes | | DNA repair ratio (SSA/NHEJ) |
|---|---|---|---|---|---|---|---|
| | DSB source (cis/trans) | Spacer (kb) | DRs (kb) | Ratio [*R*] (cargo/DR) | SSA (%) | NHEJ (%) | |
| *kmo*^DR0.2-SceI^ | I-*Sce*I (trans) | ~3.7 | ~0.2 | ~18.5 | 1.4 | 3.1 | 0.47 |
| *kmo*^DR0.7-SceI^ | I-*Sce*I (trans) | ~3.7 | ~0.7 | ~5.3 | 3.5 / 1.7 | 2.3 / 2.2 | 1.52 / 0.75 |
| *kmo*^DR1.5-AniI^ | I-*Ani*I (trans) | ~3.7 | ~1.5 | ~2.5 | 1.1 | 48.9 | 0.02 |
| *kmo*^DR0.7-nos-I-SceI^ | I-*Sce*I (cis) | ~7.9 | ~0.7 | ~11.3 | 0.5 | 1.1 | 0.45 |
| *kmo*^DR1.5-nos-I-SceI^ | I-*Sce*I (cis) | ~7.9 | ~1.5 | ~5.3 | 1.8 | 0.9 | 1.86 |

**Fig. 6 | The summary for context-dependent DSB repair processing of SSA-based transgenes at *Ae. aegypti kmo*.** Transgenic mosquito strains were generated to contain various lengths of DR motifs (0.2-kb, 0.7-kb, or 1.5-kb) and the intervening cargo genes (3.7-kb or 7.9-kb), and their DNA repair dynamics were evaluated by crossing DSB trigger strains that express homing endonucleases (I-*Sce*I or I-*Ani*I) under the control of germline-specific *nos* promoter. One-piece SSA-featured

mosquitoes were crossed with the *kmo*^−/−^ strain in DNA repair tests. The [*R*] analysis presents the proportional values of cargo lengths (bp) divided by direct repeat lengths (bp) with respect to eliminating efficiency of *kmo*-integrated transgenes. Selection biases of DNA repair pathways were shown by the proportional values of SSA rates (%) normalized by NHEJ rates (%).

phosphotransferase (HPT) and CRISPR elements were flanked by two-separated portions of the GUS reporter gene (GU and US), which are converted into the GUS gene located in the sister chromosome, resulting in marker-excised progeny (>50% at T1), though in this case portions of the introduced transgene remained behind.

As a subtype of HDR, the SSA pathway requires extensive DNA end resection (the 5′-to-3′ nucleolytic process at the DSB site), generating long 3′-single-stranded DNA (3′-ssDNA) tails[45]. A competing complex can initiate the resection of one strand at both ends of the break[49] and use the now single-stranded tails to perform a homology search for annealing between two identical sequence motifs[50]. As a critical factor, mammalian CtIP is recruited to the MRN complex and promotes the initial resection process of the broken ends by generating ssDNAs[51]. In *Drosophila*, in vivo DSB reporter assays showed that CtIP was critical for efficient SSA under short (~550 bp) and extensive (~3.6 kb) distance of resection[52], suggesting that the initial generation of ssDNA may be a rate-determining process. Similarly, loss of Marcal1, an annealing factor that binds to ssDNA–dsDNA interfaces[53,54], reduced SSA frequencies compared to wild type regardless of homology length[55].

The threshold requirements for efficient SSA are likely to vary based on host organism, chromosomal loci, lengths and arrangements of direct repeats, and resection distances[55–60]. Intrachromosomal deletions mediated by the SSA pathway can result in deletions of at least 80-kb at high efficiency[34] with the length of repeats and distance of separation strongly influencing this mode of repair in *Drosophila melanogaster*[61], *Saccharomyces cerevisiae*[34], and vertebrate cells[35]. In the yeast genome, SSA-mediated deletions occurred preferentially between the DR motifs closest to the DSB[62], and its frequency was linearly dependent on the length of DRs[57]. Moreover, it was suggested that a competition can occur between two pairs of DRs, and a preference can be made based on their relative sizes and proximity to the DSB[57]. Another assay revealed that SSA rates were enhanced when 500-bp homologies were aligned to be the annealing target farther to the DSB site than closer[55]. In this work, we hypothesized that a larger DR motif may be more favorable for SSA-based transgene elimination. However, our multi-generational tests showed that increasing DR length from 0.7-kb to 1.5-kb did not result in a difference in SSA frequencies, implying that DR length may not be a rate-limiting factor for SSA-based transgene removal in *Ae. aegypti*, at least amongst the lengths tested here. Indeed, a ratio of 18.5 (~0.2-kb DR for ~3.7-kb cargo) was shown to be sufficient for triggering SSA, implying a potential that ~1-kb length of DR sequences may trigger SSA for a larger size of gene drive (typically >15-kb). For *P5-EGFP*, only ~20-bp of DRs were able to delete >2-kb of the transgenic cargo. Our current results indicate that SSA can occur with a spacer length of approximately 8-kb in the *Aedes aegypti* genome. Overall, various sizes of DRs and cargos are feasible for genetic control approaches, and characterizing the maximum size of the transgenic cargo that can be processed by SSA may depend more on the individual target gene context.

Our DNA repair tests showed that two types of homing endonucleases, I-*Ani*I and I-*Sce*I, were associated with very different repair outcomes depending on the target transgene. In the *P5-EGFP* strain, I-*Ani*I was substantially more active than I-*Sce*I as measured by the rates of $F_2$ marker phenotypes associated with either SSA or NHEJ in $F_1$ offspring, while maintaining a relatively similar ratio between NHEJ and SSA outcomes (~1:1). In stark contrast, DSB repair at *kmo*[DR1.5-AniI] was processed almost exclusively by NHEJ. While the genomic loci containing either the *nos-I-AniI* and *nos-I-SceI* transgenes may have some differential influence on expression levels (spatial/temporal/cell type), this is insufficient to explain such strong differences in repair outcomes due to the fact that the same drivers were used for both *P5-EGFP* and *kmo*[DR] crosses. Measuring rates of SSA at more sites and with a wider array of nucleases targeting regions with varied characteristics may shed additional light on critical factors that control repair outcomes.

To date, we have exclusively used the *nanos* promoter to express the homing endonucleases that trigger SSA and transgene removal. At the same time, various CRISPR-based homing gene drive cassettes engineered in *Aedes aegypti* have been developed under the control of regulatory sequences derived from *exuperantia* (*exu*, AAEL10097), *zero population growth* (*zpg*, AAEL6726), *nanos* (*nos*, AAEL012107), *β2-tubulin* (AAEL019894), *benign gonial cell neoplasm* (*bgcn*, AAEL004117), *suppressor of defective silencing 3* (*sds3*, AAEL002084), and *nup50* (AAEL005635)[11,12,17,18,63]. Each of these exhibited differences in the level and development stage of expression, subsequently affecting repair outcomes. Among them, the gene regulatory elements of *sds3*, whose gene function is critical in the development of testis and ovarian follicles[64], was shown to lead to the highest level (~90%) of gene drive inheritance at the *kmo* locus[11]. The use of these alternative regulatory elements may also improve the selectivity of SSA over NHEJ. NHEJ-induced indels are an impediment to both homing gene drive[43,44] and SSA-based transgene removal as described here. In the current work, we screened individual founders for the emergence of indels every generation to preserve SSA-compatible progeny that maintain the intact I-*Sce*I site. Otherwise, DSB-resistant progeny may have accumulated within the test population and potentially halted the self-eliminating effect. While such an accumulation would have confounded the experiments described here, our previous mathematical modeling predicted that even with some NHEJ, low rates of SSA (~2% for maternal triggers and ~0.5% for paternal triggers) are sufficient for removing the gene drive mosquitoes from the field in ~60 generations with 5% fitness cost per transgene, or in ~35 generations if the strategy is engineered for the complete lethality cost[33]. That being said, the models suggest that further improvements to lower the NHEJ rate will increase the utility of SSA-based transgene removal approaches, for example by interfering with core NHEJ factors to bias DSB repair pathway toward homology-based repairs[46]. Thus, SSA-based self-eliminating transgenes should be further optimized with respect to potential interactions of individual components and DNA repair pathway choice in a target species.

## Methods
### Mosquito rearing
The *Aedes aegypti* Liverpool (*Lvp*) wild-type strain was maintained at 27 °C and 70% (±10%) relative humidity, with a day/night cycle of 14-h light and 10-h dark. Fertilized eggs were hatched in a pan filled with 2 L of distilled water and 200 mg of powdered fish food (TetraMin Tropical Flakes). At the L1 instar stage, larvae were manually counted to be ~500 per pan and replenished with fresh water and food every two days. Pupae were manually picked and separated with a size-based sorter and/or by identifying genitalia under the dissecting microscope. Adult mosquitoes were fed 10% sucrose solution and mated in a ♂:♀ ratio of 1:3. The mated females were fed on defibrinated sheep blood (Colorado Serum Company) using an artificial membrane feeder and oviposited on a wet-filter paper kept in a cup of ~30 ml of distilled water. The egg papers were air-dried and heat-sealed in polyethylene tubing for up to 4 months of storage at room temperature.

### Generation of transgenic strains
CRISPR/Cas9-based site-specific integrations at the *Ae. aegypti kmo* site were obtained by microinjection into pre-blastoderm embryos[65–67]. For each strain, the injection mix included 400 ng/μl of Cas9 enzyme (PNA Bio), 100 ng/μl of sgRNA, and 300 ng/μl of donor plasmid and was microinjected into pre-blastodermic embryos of a recipient strain (Supplementary Table S1). $G_0$ survivors were outcrossed with *Lvp* wild-type or *kmo*-null (*kmo*[−/−]) strain[68], and transgenic mosquitoes were screened for their fluorescent marker expression at $G_1$ larval phases. Chromosomal integration of the transgene at the *kmo* locus was confirmed by PCR analysis using genomic DNAs as the template and a primer set that is specific to both the transgene and *kmo* outside of the homology arm sequence (Supplementary Table S2).

To generate a transgenic strain expressing Y2-I-*Ani*I, donor plasmid (500 ng/μl), pMOS-3xP3-BFP-nos-I-AniI was microinjected into pre-blastoderm embryos of the *kmo*[−/−] strain, along with the *Mos1* helper

plasmid (200 ng/µl), pKhsp82M[69]. For transgenic mosquitoes, transposon-chromosome junction sequences were amplified by inverse PCR using AluI-digested genomic DNA and *Mos1* inverse repeat sequences-specific primers (Supplementary Table S2). Subsequently, the targeted chromosomal gene locus (AAEL010783 in Chr2) was identified by BLAST analysis through the VectorBase database[70].

## Subcloning

For *kmo*-targeted transgenic strains (Supplementary Table S1), donor plasmids were made by Golden Gate Assembly (NEB) with various combinations of three pieces of plasmid: (i) pGSP1 series provided homology arm 1 (HA1) and/or various sizes (0.2-kb, 0.7-kb, and 1.5-kb) of direct repeat sequences (DRs) at *kmo*; (ii) pGSP2 series provided homing endonuclease recognition sites (I-*Sce*I or Y2-I-*Ani*I) in-frame with the translational start codon (ATG) of the DsRED marker gene or the sgRNA-HybRED site for the introduction of SSA-featured DNA constructs; (iii) pGSP3 series provided HA2, the EGFP and/or BFP marker gene, and/or the *nos* promoter-driven I-*Sce*I gene. For the details, (1) pSSA-KmoDR0.2-SceI (*kmo*$^{DR0.2-SceI}$): pGSP1-KmoHA1-DR0.2 + pGSP2.3-DsRED-SV40 (SceI) + pGSP3.8 C#5-EGFP-KmoHA2; (2) pBR-KmoEx4#5 (*kmo*$^{EGFP#5}$): pGSP1-KmoHA1 + pGSP2-REDh-SV40 + pGSP3.8 C#5-EGFP-KmoHA2; (3) pSSA-KmoDR1.5-AniI (*kmo*$^{DR1.5-AniI}$): pGSP1-KmoHA1-DR1.5 + pGSP2.1-DsRED-SV40 (AniI) + pGSP3.8 C#5-EGFP-KmoHA2; (4) pBR-KmoEx4#5-nos-SceI (*kmo*$^{EGFP#5-nos-SceI}$): pGSP1-KmoHA1 + pGSP2-REDh-SV40 + pGSP3.8 C#5-EGFP-nos-SceI-KmoHA2; (5) pSSA-KmoDR0.7-nos-SceI (*kmo*$^{DR0.7-nos-SceI}$): pGSP1-KmoHA1-DR0.7 + pGSP2.3-DsRED-SV40 (SceI) + pGSP3.8 C#5-EGFP-nos-SceI-KmoHA2; (6) pSSA-KmoDR1.5-nos-SceI (*kmo*$^{DR1.5-nos-SceI}$): pGSP1-KmoHA1-DR1.5 + pGSP2.3-DsRED-SV40 (SceI) + pGSP3.8 C#5-EGFP-nos-SceI-KmoHA2.

Some assembly plasmids were generated by modifying current or previously reported ones[33]. To generate pGSP1-KmoHA1-DR0.2, the KpnI-AgeI fragment of *kmo* exons 4/5 (HA1) and *kmo* intron 3 (0.2-kb of DR) was amplified by using a primer pair of KmoHA1-F-Kpn and KmoDR0.2-R-Age (Supplementary Table S2) and replaced the counterpart regions in pGSP1-KmoHA1-DR0.7[33]. For pGSP1-KmoHA1-DR1.5, the KpnI-AgeI fragment of *kmo* exons 4/5 (HA1), *kmo* exons 2/3 (HA2, 0.6-kb of DR), and *kmo* intron 1 (0.9-kb of DR) was amplified by using a primer pair of KmoHA1-F-Kpn and In1-IR2-AgAv (Supplementary Table S2) and replaced the counterpart regions in pGSP1-KmoHA1-DR0.7[33]. pGSP2.1-DsRED-SV40 (AniI) contains the synthesized DNA piece (GenScript) that encodes DsRED with the homing endonuclease Y2-I-*Ani*I recognition sequence was engineered in-frame next to ATG translation start codon. To generate pGSP3.8 C#5-EGFP-KmoHA2, the DNA piece of PUb-EGFP-3′UTR#5 was amplified from a template plasmid pSLfa-PUb-EGFP-3′UTR#5[71] by using a primer pair of PUb-F and 3′UTR#5-R-BmBp (Supplementary Table S2). Subsequently, it replaced the counterpart regions by the flanking BlpI sites in pGSP3.8C-EGFP-KmoHA2[33]. pGSP3.8 C#5-EGFP-nos-SceI-KmoHA2 was generated by adding the DNA piece that contains 3xP3-BFP-SV40 and nos-I-SceI-3′UTR into BamHI sites in pGSP3.8 C#5-EGFP-KmoHA2.

For pMOS-3xP3-BFP-nos-AniI (*nos-I-AniI*), the DNA piece containing the *nanos* promoter, *Y2-I-AniI*, and *nanos* 3′UTR sequences was synthesized (Epoch) and inserted to MluI and XhoI sites in pM2-3xP3-BFP, a *Mariner Mos1*-based plasmid backbone. Complete sequences of plasmid constructs used to generate transgenic mosquito lines are deposited in GenBank under accession numbers PP693225-PP693229.

## The split-SSA test

Twenty adult males of individual SSA-featured strains (*kmo*$^{DR0.2-SceI}$, *kmo*$^{DR0.7-SceI}$, or *kmo*$^{DR1.5-AniI}$) were crossed with 50 nuclease-providing females (*nos-I-SceI* or *nos-I-AniI*) in triplicated assays. $F_1$ progenies were screened for marker phenotypes that were inherited from both parents (white, DsRED$^+$, and BFP$^+$ for eyes; GFP$^+$ for the body). For each replicate, 50 $F_1$ females were outcrossed with 20 *kmo*$^{-/-}$ males and individually

oviposited by the EAgaL plate method[72]. DNA repair events per individual $F_1$ females were determined by $F_2$ larval phenotypes: (i) No DSB or complete repair, *kmo*$^{DR-SceI/-}$ (White eyes; DsRED$^+$ eyes; EGFP$^+$ body); (ii) NHEJ, *kmo*$^{indel-DsRED/-}$ (White eyes; DsRED$^-$ eyes; EGFP$^+$ body); (iii) SSA, *kmo*$^{restored+/-}$ (Black eyes; DsRED$^-$ eyes; EGFP$^-$ body).

The SSA-featured *P5-EGFP* strain[36] was reciprocally crossed with nuclease-providing females (*nos-I-SceI* or *nos-I-AniI*). Parental crosses were made with 30 ♂ and 100 ♀ per cage in triplicated assays. For $F_1$ progenies (White eyes; DsRED$^+$ eyes; EGFP$^+$ body) in each replicate, 20 males and 50 females were outcrossed with 50 *kmo*$^{-/-}$ females and 20 *kmo*$^{-/-}$ males, respectively. Female mosquitoes were blood-fed two times, and all subsequent embryos were hatched and scored for $F_2$ larval phenotypes.

## The one piece-SSA test

For *kmo*-targeted transgenes carrying both direct repeats (SSA substrate) and the nuclease (SSA trigger), ~20–25 female mosquitoes (*kmo*$^{DR0.7-nos-SceI}$; *kmo*$^{DR1.5-nos-SceI}$) were outcrossed with 20 *kmo*$^{-/-}$ males in triplicated assays. For each replicate, ~500–800 offspring pupae were scored for DNA repair-associated marker phenotypes, as stated for the split-SSA test. In every generation, ~25 females per replicate that displayed intact marker phenotypes (White eyes; DsRED$^+$ eyes; EGFP$^+$ body) were further screened for silent indel mutation (an in-frame shift) in the DsRED gene by high-resolution melt analysis (HRMA). Only females with the active I-*Sce*I site were outcrossed with *kmo*$^{-/-}$ males for the offspring generation. For the single-generation assay, HRMA-verified $G_{10}$ females were allowed to individually oviposit, and progeny quantified by the EAgaL plate method[72]; $G_{11}$ progeny per female were scored for DNA repair pathway-dependent marker phenotypes.

## Sequencing analysis

To dissect the rates of individual DR motif-dependent processes within *P5-EGFP*, the split-SSA test was performed in triplicates. In $F_2$ larval screening, L2 instar larvae (*n* = 100 per replicate) were collected for WR marker phenotypes (Kmo$^-$; DsRED$^+$), which resulted from the elimination of [PUb-EGFP-SV40]. Subsequently, genomic DNA was purified from each replicate pool as PCR templates, and a primer pair that is specific to 3′-DsRED (DsRED-3F1) and the Mos arm (MLF1) (Supplementary Table S2) were used for the amplification of SSA-processed regions (~0.7-kb to 0.9-kb), which were gel-extracted for Oxford Nanopore sequencing[73]. The initially obtained raw reads (*n* = 4449) were filtered by using 'Amplicon_sorter'[74] to isolate 1783 reads (40.1%) greater than or equal to the 0.5-kb cutoff and sort them into groups based on sequence similarity. These consensus groups were then aligned to seven types of signature sequences (~60–100 nucleotides), which represent the predicted end-products of SSA-based DSB processes occurred within *P5-EGFP*, by using Minimap2[75] enforcing a mapping quality of 20, with potential secondary alignments further filtered to obtain the successful alignment of 794 reads. Lastly, these alignments were sorted and indexed using SAMtools[76] for subsequent visualization and analysis via the Broad Institute Integrative Genomics Viewer (IGV) web application[77]. Signature sequences of DSB repair end-products in *P5-EGFP* (FASTA) are described in Supplementary Note 2.

## Statistics and reproducibility

As indicated, comparisons were performed ANOVA followed by either Tukey's multiple comparison test (if data were normal) or the Kruskal–Wallis test (if normality test was failed) as implemented in Graphpad Prism V10.2.2. Thresholds for significance are described in each figure legend. All experiments were repeated a minimum of three times, number of mosquitoes screened was as large as possible given the realities of space and personnel available. Number of progeny scored was only limited by the reproductive capacity of each female (ie, all progeny were scored). Number of individuals genotyped (50–100) again as a compromise between

depth and available resources. Replicates were either at the level of a pooled cross (all progeny from a group of founders) or from independent founders (when progeny were scored individually).

## Reporting summary
Further information on research design is available in the Nature Portfolio Reporting Summary linked to this article.

## Data availability
Plasmid sequences are deposited in GenBank (PP693225-PP693229) and amplicon sequencing data are available at NCBI Sequence Read Archive under BioProject ID PRJNA1101529. Raw data is available in Supplementary Data; uncropped images are available in Supplementary Fig. 5. All other data are available from the corresponding author (or other sources, as applicable) on reasonable request.

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

## Acknowledgements

We thank the research staff in the Adelman laboratory at Texas A&M University for excellent technical assistance for this study. This work was supported by the National Institute of Allergies and Infectious Diseases of the National Institutes of Health (NIH-NIAID) under award numbers (R01AI148787, R01AI137112). The content is solely the responsibility of the authors and does not necessarily represent the official views of the National Institutes of Health.

## Author contributions

K.C. and Z.N.A. conceived the research and experimental design and wrote the manuscript. K.M.M. revised the manuscript. K.C. conducted the experiments. B.C. performed assays of transgene-eliminating dynamics and phenotypic screening. J.S.R. provided mass sequence alignment analysis.

C.D. participated in the generation of transgenic strains. All authors read and approved the final manuscript.

## Competing interests

Z.N.A. and K.M.M. are listed as inventors on provisional patent TAMC:054USP2 related to self-eliminating transgenes.
