## [Peer review file · Communications Biology]

Reviewers' comments:

Reviewer #1 (Remarks to the Author):

In their manuscript entitled "Transgene removal using an in cis programmed homing endonuclease and the impact of direct repeat length, nuclease and genomic context on single-strand annealing in the mosquito *Aedes aegypti*," Chae and colleagues clearly present a body of work and a manuscript that adds to the genetic toolbox for the yellow fever mosquito and also points to some parameters (e.g. repeat length and genomic context) that can help explain variable success of genomic editing in mosquitoes using these tools.

The method in question, transgene removal using a programmed endonuclease and repair via single-strand annealing (SSA), is relevant to the gene drive field because it has been floated as a potential pre-programmed strategy to render a gene drive ineffective after a certain period of time. This manuscript demonstrates the removal of an ~8kb fragment using this approach - a first of its kind in *Aedes aegypti* - and points to some parameters (e.g. nuclease selection, Directed Repeat length, genomic context) that appear critical in biasing SSA over NHEJ.

In general, the figures are clearly laid out, the data well presented and the experiments well controlled, and the supplements are clear and comprehensive. In total, the manuscript overall is an excellent addition to the body of knowledge surrounding gene editing in *Aedes aegypti*. In particular, I found the results section to be clearly written especially given the potentially complex combinatorial outcomes of each experiment. The body of work is solid and in this reviewer's opinion should be published without asking for additional major experimental evidence, which likely lies beyond the timeline and scope of this particular paper.

Major comments:

1) my major comment is a frustration shared no doubt by the authors themselves, which is that developing and testing gene editing techniques and approaches *Aedes aegypti* appears to be resistant to logic and extrapolation from foundational gene editing results in other systems e.g. yeast or *Drosophila*. In particular, the dramatic impact of genomic locus/context on repair efficiency and outcomes is particularly vexing. While this may be a difficult proposal to implement, as a reader I would appreciate any speculation in the discussion (preferably based on informatics analyses of the P5 and kmo loci and/or data on these nucleases activity from other systems) as to what specifically might be influencing these results: chromatin conformation? Sequence parameters at or around the cut site (e.g. %GC or repeat content)?

2) The analysis of the nanopore data and corresponding data analysis should be more fully explained. The authors appear to have aligned the reads to a database of 7 predicted outcomes, resulting in the

pie chart in Figure 4. My reading of the methods suggests that there were in fact 1783 reads that passed initial QC cutoffs and were then grouped into 'consensus groups' — more detail is needed here. Were there 991 consensus groups? Or more likely, is there a mismatch with some reads passing QC but not aligning to the 7 predicted outcomes? What did those look like? This is not necessarily a quibble with the methods used, but an ask for more detail and clarity given the potential ability of the long-read sequencing to uncover potentially unexpected results rather than simply counting instances of 7 predicted outcomes.

Minor comments

3) one major caveat when thinking about the use of this technology in deployed gene drives is the potential generation of mutations refractory to nuclease activity and desired outcomes. This is indicated by the authors themselves, who had to screen for I-SceI target site completeness prior to considering an individual as a potential founder. This should be raised in the discussion

4) Figure 3d - were these sequences generated directly from the HRMA data, or from Sanger sequencing/other methods? This should be clarified (and apologies if I missed it in the methods)

5) Line 255: "regardless of spacer length" — this should be clarified. given the yeast results stated above that SSA can close 80kb gaps, but in mosquitoes, only much smaller gaps were tested.

6) I also find the mock calculation on lines 252-253 confusing - are the authors proposing a linear scaling of DR size with deletion size efficiency, which seems a bit dodgy as a hypothesis based on extrapolation from only a few datapoints?

Signed, Ben Matthews

Reviewer #2 (Remarks to the Author):

Dr. Adelman and colleagues present a manuscript in which they make an exciting advancement in the development of gene-drive for mosquito-borne disease control. The authors detail in vivo assay of transgene elimination by transgene-encoded endonuclease and single-strand annealing. Mosquitoes are the most immediately relevant organisms for studying gene drive mechanisms on the cellular and molecular levels, as they are the very near real-world application. So, this work not only informs mosquito biology and efforts, but also is important broadly in fields where gene-drive could be used in the future. This includes public health, conservation and agricultural biology fields. An important

outstanding question is whether and how a synthetic transgene can be designed to remove itself from a population after it has been spread via gene drive. The authors build off of their own previous genetic engineering work to show that indeed, using cis-encoded endonucleases, removal of a transgene can occur by single-stranded annealing, provide both the first demonstration of feasibility of self-reversible transgenes that restore the original genotype as well as beginning to define the operation parameters that will be necessary to use such a design in a disease control application.

In our discipline, this set of experiments is outstanding in its execution. It is extremely challenging to generate transgenic mosquito lines, but this group has presented a thorough characterization of seven transgenic lines and conducted a rigorous set of assays using these lines. This is an excellent contribution and necessary to ask their questions, but again, challenging and time consuming-15,000 mosquito embryo injections alone is a laudable.

Overall, the authors are able to use their assays to draw several important and well-founded conclusions about SSA-based transgene self-elimination. First, that a relatively large transgene cargo can be removed using a relatively small direct repeat means is an encouraging finding that this will be usable in transgenes that will have multiple gene functions. That different endonucleases behave differently indifferent genomic and transgene contexts is an important finding; it will be necessary to characterize the activity of specific endonucleases with each transgene design in the future. Multiple repair pathways in addition to SSA will repair the cuts induced by endonucleases, that a substantial portion of double-stranded breaks are repaired by NHEJ instead of SSA is important; follow up design will be necessary to build systems in which NHEJ-repair does not compromise removal of the transgene.

This manuscript is well-written, with extremely thorough description of methods, good visual presentation of data and I see no technical or conceptual flaws that should preclude its publication in your journal.

Reviewer #3 (Remarks to the Author):

In this manuscript, Chae and colleagues utilize elegant genetics assays to induce DNA double-strand breaks (DSBs) followed by repair by either single-strand annealing (SSA) or non-homologous end-joining (NHEJ) in *Aedes aegypti*. Their goal was to determine whether the source of the DSB-induction (homing endonuclease), length of the repetitive sequences, or genomic context impacts repair by SSA or NHEJ. Long-term goals of using gene drive to control mosquito vector populations would potentially require error-prone SSA or NHEJ to disrupt gene expression in natural populations, thus understanding the implications of these factors in repair is critical.

Briefly, this group designed an array of DSB repair reporter cassettes with variations in either homing endonuclease site, length of direct repeats, genomic loci, or whether endonuclease expression was in cis or in trans. Genetic cross schemes produced females experiencing break events in the germline. These females were crossed to tester males, and using phenotypic analyses of their progeny, the authors determined which individual females were capable of repairing by either SSA or NHEJ as well as the frequency of these individual repair events. Molecular analyses using Oxford Nanopore sequencing provided sequence data describing the NHEJ indels at the site of the breaks from various constructs. The authors found that repeat length increases SSA frequency, I-Anil induces break repair more frequently than I-SceI, but cassette size does not (generally) impact repair. Importantly, the authors demonstrated the ability for transgene elimination in cis, which is critical for practical applications.

This is a well-written paper with clear figures to follow the complex data sets and molecular outcomes that emerge from these analyses. Application of these elegant genetic systems has significant implications in utilizing gene drives for insect control. This study underscores the variability of repair outcomes in multicellular organisms which has wide implications to the field of gene drive.

A few clarifying questions and/or minor suggestions to consider (particularly for those not well-versed in *Aedes* genetics and molecular outcomes of DSB repair events):

1) Figure 1a, it may be helpful to illustrate the repair events after repair. For example, what would an SSA event look like? Illustrating the loss of the intervening sequences may be helpful for those less familiar with the molecular outcomes of SSA. Similar to an NHEJ event, demonstrating how the phenotypic markers will persist could add clarity.

2) Figure 1 legend, ** is defined as significance of $P < 0.005$. Should this be $P < 0.01$, which is aligned with a more conventional report of statistics? Related, Figure 4, the asterisks may be modified to follow the more conventional representations: * $p < 0.05$, ** $p < 0.01$, *** $p < 0.001$, **** $p < 0.0001$, etc.

3) Figure 2c, are the differences in NHEJ or SSA statistically significant across the generations? If not, this should be added to the figure and/or the figure legend.

4) Figure 3d, do the colors in the individual sequence events correspond to the groups in panel b? Please add clarifying language to the figure legend.

5) Figure 5a, is the I-SceI cassette is the same as in Fig. 1A?

6) Figure 6- reminding the reader that this is a summary of the data from the rest of the manuscript with the additional “R” analysis would be helpful.

7) Data demonstrating that transgene elimination through SSA in cis is quite important to downstream applications. Is it possible to predict how many generations it would take to completely eliminate the transgene in a population? This could be a helpful discussion point to expand on this important finding.

8) References to consider within the discussion: line 246, papers by Yannuzzi et al (<https://www.ncbi.nlm.nih.gov/pmc/articles/PMC8468788/>) and Dewey et al (<https://www.ncbi.nlm.nih.gov/pmc/articles/PMC9836020/>) more appropriately demonstrate how the length of the DR in *Drosophila* impact SSA frequencies. Similarly, Sugawara et al describe how DR length impacts SSA in *S cerevisiae* (<https://www.ncbi.nlm.nih.gov/pmc/articles/PMC85979/>).

9) Minor typos:

a. Line 100, “intron1” should be intron 1

b. Line 575, “note that founders that produced both types of events” (“both” should be added, I believe?)

Reviewers' comments:

Reviewer #1 (Remarks to the Author):

In their manuscript entitled “Transgene removal using an in cis programmed homing endonuclease and the impact of direct repeat length, nuclease and genomic context on single-strand annealing in the mosquito *Aedes aegypti*,” Chae and colleagues clearly present a body of work and a manuscript that adds to the genetic toolbox for the yellow fever mosquito and also points to some parameters (e.g. repeat length and genomic context) that can help explain variable success of genomic editing in mosquitoes using these tools.

The method in question, transgene removal using a programmed endonuclease and repair via single-strand annealing (SSA), is relevant to the gene drive field because it has been floated as a potential pre-programmed strategy to render a gene drive ineffective after a certain period of time. This manuscript demonstrates the removal of an ~8kb fragment using this approach - a first of its kind in *Aedes aegypti* - and points to some parameters (e.g. nuclease selection, Directed Repeat length, genomic context) that appear critical in biasing SSA over NHEJ.

In general, the figures are clearly laid out, the data well presented and the experiments well controlled, and the supplements are clear and comprehensive. In total, the manuscript overall is an excellent addition to the body of knowledge surrounding gene editing in *Aedes aegypti*. In particular, I found the results section to be clearly written especially given the potentially complex combinatorial outcomes of each experiment. The body of work is solid and in this reviewer’s opinion should be published without asking for additional major experimental evidence, which likely lies beyond the timeline and scope of this particular paper.

Major comments:

1) my major comment is a frustration shared no doubt by the authors themselves, which is that developing and testing gene editing techniques and approaches *Aedes aegypti* appears to be resistant to logic and extrapolation from foundational gene editing results in other systems e.g. yeast or *Drosophila*. In particular, the dramatic impact of genomic locus/context on repair efficiency and outcomes is particularly vexing. While this may be a difficult proposal to implement, as a reader I would appreciate any speculation in the discussion (preferably based on informatics analyses of the P5 and kmo loci and/or data on these nucleases activity from other systems) as to what specifically might be influencing these results: chromatin conformation? Sequence parameters at or around the cut site (e.g. %GC or repeat content)?

***Response:** We certainly share the reviewer’s sense of wonder here. With only two genomic positions at play here, we resisted the temptation to speculate as to specific features that might skew outcomes. Though the reviewer’s invitation to do so is tempting, we simply do not have enough information to even begin to guess. We are in the process of targeting the same transgene with many different nucleases (CRISPR-based, so lots of targets), so hopefully that will shed some additional light on important sequence characteristics. This is now mentioned in the manuscript.*

2) The analysis of the nanopore data and corresponding data analysis should be more fully explained. The authors appear to have aligned the reads to a database of 7 predicted outcomes, resulting in the pie chart in Figure 4. My reading of the methods suggests that there were in fact 1783 reads that passed initial QC cutoffs and were then grouped into ‘consensus groups’ — more detail is needed here. Were there 991 consensus groups? Or more likely, is there a mismatch with some reads passing QC but not aligning to the 7 predicted outcomes? What did those look like? This is not necessarily a quibble with the methods

used, but an ask for more detail and clarity given the potential ability of the long-read sequencing to uncover potentially unexpected results rather than simply counting instances of 7 predicted outcomes.

Response: We have added more detail to the methods to clarify this as requested by the reviewer. In short, we used several approaches (other mapping programs, de novo clustering, etc...) to try to extract unbiased repair outcomes from the remaining reads but were unable to do so. This is largely due to the high error rate of individual nanopore reads, where error rates remain above 10% and the initial 500bp cutoff still resulting in many reads being too short to provide useful information (amplicon size was 900bp). We added the text below as "Supplemental Text 1" to the manuscript.

Analyzing 'amplicon_sorter' groups and unaligned reads directly

'Amplicon_sorter' filtered raw .fastq reads by size (0.5-kb) and binned reads into groups based on sequence similarity - referred to as 'consensus groups' in the paper. The purpose of read binning and creation of 'consensus groups' was to detect all editing outcomes regardless of alignment. Reads were binned into consensus groups if they were 80% similar, if not they were binned into unique groups. In total, 4 'consensus groups' were formed (n= 1361/1783 total reads) and 1 'unique' group was formed that did not meet the 80% cutoff criteria (n= 422/1783 total reads). These reads were retained as .fastq files and aligned using minimap2 to exclude secondary alignments and alignments with mapping qualities less than 20 in order to increase confidence in alignments (note: this decreased initial total alignments reported from 991/1783 to 794/1783 total reads). Thus, with our new alignment parameters, 989/1783 reads did not align to signature sequences, but met the 0.5-kb cutoff filter.

Further interrogation of reads contained within the 'unique' groups from 'amplicon_sorter' were aligned using Seqman Ultra to the entire P5 transgene and revealed 1,008 reads contained that did not contain the primer landing sites, but spanned the DsRED open reading frame and SV40 sequence part of the intended amplicon. It is possible these reads could be error-prone PCR amplicons or incomplete linear nanopore reads, in which case even small amounts of mismatches or shorter truncations present in raw reads may result in unsuccessful alignment to the 7 signature sequences. Further, to bypass potential errors from 'amplicon_sorter', we analyzed unaligned reads contained within the .sam file output from minimap2, used 'fastq-filter' to isolate 0.5-kb or larger reads (1783 total reads remained), and performed a new alignment using the entire P5 transgene as a reference and minimap2 under higher alignment sensitivity parameters (namely, by decreasing the minimizer k-mer seed length from 15 to 5 to allow more alignments with shorter exact matches). After visualizing on IGV, 109 aligned reads spanning the DsRED open reading frame and SV40 sequence were recovered. Though fewer reads were aligned likely due to the alignment stringency of minimap2 even with manual changes to the alignments parameters, the consistency in the mapping location of these alignments by minimap2 and Seqman Ultra adds confidence that these reads are either incomplete or of lower quality, and as a result were either unaligned or filtered out during fastq quality control steps.

Increasing alignment sensitivity to catch unaligned reads

Next, sensitivity of minimap2 alignment was increased by decreasing the minimizer k-mer seed length from 15 to 10 to catch previously unaligned raw reads - these changes in alignment parameters decreased the minimum number of exact matches required for alignment initialization and allowed for less-specific alignments. Total aligned reads only modestly increased from 794 to 831 and SSA repair product frequencies remained nearly identical (59.32% by SV40, 34.06% by I-Cmoel, 3.61% by I-AniI, 3.01 % by loxP, 0% by I-SceI, 0% by I-PpoI, and 0% by I-CreI). To further increase sensitivity, in addition to decreased minimizer k-mer seed length, previous mapping quality score filters were removed entirely. As a result, 1294 total reads aligned. Noticeable decreases in alignment specificity were observed by an increase in overall supplementary alignments and shorter/truncated alignments. Importantly, the increase in overall alignments did not change the relative frequency of SSA repair events (49% by SV40, 41% by I-Cmoel, 3.17% by I-AniI, 2.93% by loxP, 2.32% by I-SceI, 0.62% by I-PpoI, and

0.23% by I-CreI). Taken together, these unaligned reads can be recovered, but seemingly at the expense of alignment specificity.

Minor comments

3) one major caveat when thinking about the use of this technology in deployed gene drives is the potential generation of mutations refractory to nuclease activity and desired outcomes. This is indicated by the authors themselves, who had to screen for I-SceI target site completeness prior to considering an individual as a potential founder. This should be raised in the discussion.

Response: This has been added to the discussion as suggested.

4) Figure 3d - were these sequences generated directly from the HRMA data, or from Sanger sequencing/other methods? This should be clarified (and apologies if I missed it in the methods).

Response: Those sequences were obtained by Sanger sequencing of the HRMA samples in Figure 3c. The colors were matched between HRMA peaks in Figure 3c and IDs in Figure 3d. We have updated the legend “d Sequence analysis of amplicons with altered melting curves (Fig. 3c), compared to the baseline. Sequencing analysis revealed various numbers of nucleotide deletions occurred at site-A. The colors of IDs designate individual HRMA peaks shown in Figure 3c.”

5) Line 255: “regardless of spacer length” — this should be clarified. given the yeast results stated above that SSA can close 80kb gaps, but in mosquitoes, only much smaller gaps were tested.

Response: We have updated the sentence, where the interpretation is limited to our results from this study: “Our current results indicate that SSA can occur with a spacer length of approximately 8-kb in the Aedes aegypti genome.”

6) I also find the mock calculation on lines 252-253 confusing - are the authors proposing a linear scaling of DR size with deletion size efficiency, which seems a bit dodgy as a hypothesis based on extrapolation from only a few datapoints?

Response: The text was updated to be a general prediction: Indeed, R18.5 (~0.2-kb DR; ~3.7-kb cargo) was shown to be sufficient for triggering SSA, implying a potential that ~1-kb length of DR sequences may trigger SSA for a larger size of gene drive (>15-kb).

Signed, Ben Matthews

Reviewer #2 (Remarks to the Author):

Dr. Adelman and colleagues present a manuscript in which they make an exciting advancement in the development of gene-drive for mosquito-borne disease control. The authors detail in vivo assay of transgene elimination by transgene-encoded endonuclease and single-strand annealing. Mosquitoes are the most immediately relevant organisms for studying gene drive mechanisms on the cellular and molecular levels, as they are the very near real-world application. So, this work not only informs mosquito biology and efforts, but also is important broadly in fields where gene-drive could be used in the future. This includes public health, conservation and agricultural biology fields. An important outstanding question is whether and how a synthetic transgene can be designed to remove itself from a population after it has been spread via gene drive. The authors build off of their own previous genetic engineering work to show that indeed, using cis-encoded endonucleases, removal of a transgene can occur

by single-stranded annealing, provide both the first demonstration of feasibility of self-reversible transgenes that restore the original genotype as well as beginning to define the operation parameters that will be necessary to use such a design in a disease control application.

In our discipline, this set of experiments is outstanding in its execution. It is extremely challenging to generate transgenic mosquito lines, but this group has presented a thorough characterization of seven transgenic lines and conducted a rigorous set of assays using these lines. This is an excellent contribution and necessary to ask their questions, but again, challenging and time consuming-15,000 mosquito embryo injections alone is a laudable.

Overall, the authors are able to use their assays to draw several important and well-founded conclusions about SSA-based transgene self-elimination. First, that a relatively large transgene cargo can be removed using a relatively small direct repeat means is an encouraging finding that this will be usable in transgenes that will have multiple gene functions. That different endonucleases behave differently indifferent genomic and transgene contexts is an important finding; it will be necessary to characterize the activity of specific endonucleases with each transgene design in the future. Multiple repair pathways in addition to SSA will repair the cuts induced by endonucleases, that a substantial portion of double-stranded breaks are repaired by NHEJ instead of SSA is important; follow up design will be necessary to build systems in which NHEJ-repair does not compromise removal of the transgene.

Response: From line 310, we discuss SSA-based transgene self-elimination further based on the previous modeling works in regard to the predicted performance of self-eliminating gene drive and the emergence of NHEJ-driven resistant alleles.

This manuscript is well-written, with extremely thorough description of methods, good visual presentation of data and I see no technical or conceptual flaws that should preclude its publication in your journal.

Reviewer #3 (Remarks to the Author):

In this manuscript, Chae and colleagues utilize elegant genetics assays to induce DNA double-strand breaks (DSBs) followed by repair by either single-strand annealing (SSA) or non-homologous end-joining (NHEJ) in *Aedes aegypti*. Their goal was to determine whether the source of the DSB-induction (homing endonuclease), length of the repetitive sequences, or genomic context impacts repair by SSA or NHEJ. Long-term goals of using gene drive to control mosquito vector populations would potentially require error-prone SSA or NHEJ to disrupt gene expression in natural populations, thus understanding the implications of these factors in repair is critical.

Briefly, this group designed an array of DSB repair reporter cassettes with variations in either homing endonuclease site, length of direct repeats, genomic loci, or whether endonuclease expression was in cis or in trans. Genetic cross schemes produced females experiencing break events in the germline. These females were crossed to tester males, and using phenotypic analyses of their progeny, the authors determined which individual females were capable of repairing by either SSA or NHEJ as well as the frequency of these individual repair events. Molecular analyses using Oxford Nanopore sequencing provided sequence data describing the NHEJ indels at the site of the breaks from various constructs. The authors found that repeat length increases SSA frequency, I-AniI induces break repair more frequently than I-SceI, but cassette size does not (generally) impact repair. Importantly, the authors demonstrated the ability for transgene elimination in cis, which is critical for practical applications.

This is a well-written paper with clear figures to follow the complex data sets and molecular outcomes that emerge from these analyses. Application of these elegant genetic systems has significant implications in utilizing gene drives for insect control. This study underscores the variability of repair outcomes in multicellular organisms which has wide implications to the field of gene drive.

A few clarifying questions and/or minor suggestions to consider (particularly for those not well-versed in Aedes genetics and molecular outcomes of DSB repair events):

1) Figure 1a, it may be helpful to illustrate the repair events after repair. For example, what would an SSA event look like? Illustrating the loss of the intervening sequences may be helpful for those less familiar with the molecular outcomes of SSA. Similar to an NHEJ event, demonstrating how the phenotypic markers will persist could add clarity.

Response: Figure 1a and legend were updated to illustrate the repair outcomes by either NHEJ or SSA.

2) Figure 1 legend, ** is defined as significance of $P < 0.005$. Should this be $P < 0.01$, which is aligned with a more conventional report of statistics? Related, Figure 4, the asterisks may be modified to follow the more conventional representations: * $p < 0.05$, ** $p < 0.01$, *** $p < 0.001$, **** $p < 0.0001$, etc.

Response: These were corrected as advised.

3) Figure 2c, are the differences in NHEJ or SSA statistically significant across the generations? If not, this should be added to the figure and/or the figure legend.

Response: Both figure and legend were updated to include statistical analysis.

4) Figure 3d, do the colors in the individual sequence events correspond to the groups in panel b? Please add clarifying language to the figure legend.

Response: No. The individual colors of the sequences in Figure 3d originated from those of HRMA peaks in panel c. The legend of Figure 3d was updated to clarify it accordingly.

5) Figure 5a, is the I-SceI cassette is the same as in Fig. 1A?

Response: Yes. The legend was updated: Schematic representation of pSSA-KmoDR0.7-SceI (Fig. 1a).

6) Figure 6- reminding the reader that this is a summary of the data from the rest of the manuscript with the additional “R” analysis would be helpful.

Response: The legend of Figure 6 was updated as suggested.

7) Data demonstrating that transgene elimination through SSA in cis is quite important to downstream applications. Is it possible to predict how many generations it would take to completely eliminate the transgene in a population? This could be a helpful discussion point to expand on this important finding.

Response: From line 310, we discuss SSA-based transgene self-elimination further based on the previous modeling works in regard to the predicted performance of self-eliminating gene drive and the emergence of NHEJ-driven resistant alleles.

8) References to consider within the discussion: line 246, papers by Yannuzzi et al (<https://www.ncbi.nlm.nih.gov/pmc/articles/PMC8468788/>) and Dewey et al

(<https://www.ncbi.nlm.nih.gov/pmc/articles/PMC9836020/>) more appropriately demonstrate how the length of the DR in *Drosophila* impact SSA frequencies. Similarly, Sugawara et al describe how DR length impacts SSA in *S cerevisiae* (<https://www.ncbi.nlm.nih.gov/pmc/articles/PMC85979/>).

Response: Based on the suggested references, SSA dynamics were further discussed from line 263.

9) Minor typos:

a. Line 100, “intron1” should be intron 1

Response: All introns and exons were corrected consistently throughout the text.

b. Line 575, “note that founders that produced both types of events” (“both” should be added, I believe?)

Response: All corresponding texts were corrected as suggested.

REVIEWERS' COMMENTS:

Reviewer #1 (Remarks to the Author):

Having read the revised manuscript and Response to Reviewers, I feel that the authors have done a commendable job in incorporating reviewer feedback and I have no further queries.

Reviewer #3 (Remarks to the Author):

The authors have adequately addressed my concerns and/or suggestions. I see no further revisions needed for this manuscript.